



# Persistent high PM pollution in the Eastern Mediterranean and Middle East: Insights from long-term observations and source apportionment in Cyprus

Elie Bimenyimana[1], Jean Sciare[1], Michael Pikridas[1], Konstantina Oikonomou[1], Minas Iakovides[1], Emily Vasiliadou[3], Chrysanthos Savvides[3], and Nikos Mihalopoulos[1,2]

[1]Climate and Atmosphere Research Centre (CARE-C), The Cyprus Institute, Nicosia, Cyprus.
[2]National Observatory of Athens, Athens, Greece.
[3]Department of Labour Inspection, Ministry of Labour and Social Insurance, Nicosia, Cyprus.

*Correspondence to*: E. Bimenyimana (e.bimenyimana@cyi.ac.cy) and N. Mihalopoulos (n.mihalopoulos@cyi.ac.cy)

**Abstract.** Long-term daily $PM_{2.5}$ and $PM_{10}$ chemical speciation data was collected continuously from 2015 to 2023 at an urban traffic and regional background site in Cyprus, offering a unique opportunity to quantify the influence and trends of i) local emissions on urban PM concentration levels and sources, and ii) regional PM emissions over the Eastern Mediterranean basin. Despite a statistically significant drop in $PM_{2.5}$ and $PM_{10}$ at both sites over the last 19 years (2005-2023), concentration levels remain high with no further significant improvements observed over the last 9 years; making PM concentration levels well above the new EU annual limits. To refine this analysis, long-term trends (2015-2023) were explored for individual PM chemical species and sources derived by PMF source apportionment. A decreasing trend in traffic-related $PM_{10}$ of 35% was observed at the traffic site, suggesting the effectiveness of the gradual shift of the vehicle fleet towards the latest EURO-standard vehicles. On the other hand, this reduction in tailpipe traffic emissions was completely offset by an increase of uncontrolled urban emissions, such as road dust re-suspension and biomass burning from domestic heating, calling for the rapid implementation of abatement measures.

Based on cluster analysis of air mass origins, the Middle East region was identified as a major hotspot of $PM_{10}$ over the Eastern Mediterranean; with both high concentration levels of dust from the Arabian desert and substantial anthropogenic pollution with continuously increasing trends in biomass burning and sulfate-rich emissions from fossil fuel combustion over the past decade.

## 1 Introduction

Particulate matter (PM) is a key component of air pollution with profound implications for human health (EEA, 2020; Lelieveld et al., 2019; WHO, 2022) and climate (IPCC, 2021; Myhre et al., 2013). Understanding the long-term evolution of PM concentration levels and sources over time enables the evaluation of the effectiveness of mitigation policies and possibly the identification of areas for improvement.

While long-term PM trends have been extensively studied in northern and central Europe, as well as the western Mediterranean (e.g. Borlaza et al., 2022; Pandolfi et al., 2016; Querol et al., 2014), such long-term PM data remain scarce at urban and regional background sites of the Eastern Mediterranean and the Middle East (EMME) region.

Declining trend of sulfate aerosols, as recorded from long-term PM chemical composition observations over Greece, has been identified as one of the key drivers responsible for the recent warming acceleration of the Mediterranean Basin (Urdiales-Flores et al., 2023). Combining ground-based $PM_{10}$ and satellite products (dust aerosol optical depth), Achilleos et al. (2020) assessed the trends of desert dust storm across the Eastern Mediterranean basin for a 12-year period (2006-2017) both in terms of intensity and frequency but could not derive clear trends from this major regional source over this period.



The few studies in urban environments of the EMME were mostly performed in Greece and have reported a reduction in PM
concentrations, particularly before 2011, primarily resulting from policies targeting traffic emissions, followed by a rise driven by increasing residential wood burning emissions (Diapouli et al., 2017b; Kaskaoutis et al., 2023; Paraskevopoulou et al., 2015). Reduction in PM traffic emissions (by ca. 50%) was further confirmed by Gratsea et al. (2017) for the period 2000-2015, based on CO observations taken in Athens during the morning traffic rush hours. However, it is worth stating that regional and urban PM trend observations from Greece may not be fully applicable to the entire Eastern Mediterranean, given
the significant differences in terms of air masses origin (Middle East air masses are reaching the Levantine Basin and Cyprus but not Greece; Achilleos et al., 2020).

The most comprehensive long-term PM trend study conducted so far over Cyprus (located at ca. 1,000 km south-east of the Greek mainland), was performed by Pikridas et al. (2018) for a 17-year period (from 1998 to 2015) at several remote and urban locations of the national air quality monitoring network. This study reported downward trends in urban and regional $PM_{2.5}$ and
$PM_{10}$ concentrations, but was unable to draw clear conclusions on the factors driving the observed decline due to lack of information on long-term PM chemical composition and sources.

Overall, the current lack of long-term detailed PM chemical data and sources in the region constitutes a major limitation to our understanding of the PM pollution long-term dynamics in Cyprus and the EMME, a region highly affected by a wide range of regional (both natural and anthropogenic) PM pollution sources (Fadel et al., 2023; Kanakidou et al., 2011; Lelieveld et al.,
2002) further exacerbated by local uncontrolled emissions and climate change-induced environmental degradation, including prolonged droughts and severe heatwaves (Lelieveld et al., 2014; Zittis et al., 2022).

The work presented herewith stands as the most extensive long-term $PM_{2.5}$ and $PM_{10}$ chemical characterization conducted to date in Cyprus. By integrating continuous long-term (2015-2023) daily (24h) PM chemical composition and robust PM source
apportionment at both an urban traffic and a regional background site, this study reports, for the first time (to the best of our knowledge) simultaneous long-term trends in PM chemical composition and sources at both local (urban) and regional scales over the EMME.

## 2 Materials and methods

### 2.1 Sampling sites and sample collection

This study was conducted at two contrasted sites of the Cyprus Air Quality monitoring network operated by the Department of Labour Inspection (Ministry of Labour and Social Insurance); an urban traffic site (NICosia TRAffic; NICTRA) and a regional background (Agia Marina Xyliatou; AMX). These two stations are those reporting annually to the European Commission the PM chemical composition in compliance with the EU Air Quality Directive 2008/50/EC.

The NICTRA station (35°09′07″ N, 33°20′52″ E; 176 m ASL) is situated in a densely populated area within the urban
agglomeration of Nicosia, the capital city of Cyprus, and near a major urban road. The rationale for selecting this site was to capture both the impact of road traffic emissions, but also any other type of urban emissions. The AMX station (35°02′17″N, 33°03′28″ E; 532 m ASL) is representative for the regional background conditions of the Eastern Mediterranean and the Middle East, as it is located at a remote rural area, in the middle of the island, with minimal influence from local pollution. A detailed description of these two sites is available elsewhere (Bimenyimana et al., 2024; Pikridas et al., 2018; Tsagkaraki et al., 2021;
Vrekoussis et al., 2022).

Continuous (24/7) PM filter sampling was carried out concurrently at the urban traffic site (NICTRA) in $PM_{10}$ and at the regional background (AMX) in both $PM_{2.5}$ and $PM_{10}$. The sampling period extended for a 19-year period (2005-2023) for PM



gravimetry and 9-year period (2015-2023) for chemical analysis of ions, carbon and trace metal content. More specifically, $PM_{2.5}$ and $PM_{10}$ were collected on a daily basis (from midnight to midnight, Local Standard Time) onto pre-weighed 47-mm and 150-mm diameter filters (Whatman Cellulose 7194-004 from 2005 to 2008; Whatman Quartz 1851-150 for 2008-2015; Pall Tissuquartz 2500 QAT-UP afterwards) using autonomous low- and high-volume samplers (Leckel SEQ 47/50 and Digitel DHA-80), with operational flow rates of 2.3 m$^3$/h and 30 m$^3$/h, respectively. Field blanks, collected and handled in the same way as field samples, were used for blank correction.

### 2.2 PM mass determination

Filter-based PM mass concentrations presented in this work were gravimetrically determined following rigorously the CEN 12341 reference filter weighing protocol (EN 12341, 2014). More specifically, PM filter samples, along with field blanks, were conditioned for 48 hours at a room temperature of $20 \pm 1°C$ and $50 \pm 5\%$ relative humidity. Then, they were weighed using a 6-digit analytical microbalance (Mettler Toledo, Model XP26C), prior to and after filter sampling.

Due to technical issues encountered with the $PM_{2.5}$ filter weighing process at AMX for the period 2011-2014, the gravimetric $PM_{2.5}$ data was substituted with daily averaged online measurements from a co-located TEOM instrument equipped with a Filter Dynamics Measurement System (FDMS; Thermo model 1405DF). Note that a good agreement ($r^2$=0.96; slope=1.02) between TEOM-FDMS and gravimetric PM mass determination methods has been reported by Pikridas et al. (2018) at that location.

### 2.3 PM chemical analyses

Filter samples were analyzed for a range of PM chemical components, including major ions, carbonaceous species (Organic carbon, OC, and elemental carbon, EC) and trace metal elements, using standard analytical methods, as described below.

Major cations ($Na^+$, $K^+$, $NH_4^+$, $Mg^{2+}$, and $Ca^{2+}$) and anions ($Cl^-$, $NO_3^-$, $SO_4^{2-}$) were quantified by Ion Chromatography (IC; Thermo Scientific, Model ICS-5000) as described in details in Sciare et al. (2005). A filter punch was extracted with ultrapure water for 45 min by sonication. With this extraction method, over 98% recovery is achieved for all targeted species. Cations and anions were separated using MSA and KOH as eluents in isocratic and gradient mode, respectively.

Organic and Elemental Carbon (OC and EC) analysis was conducted on filter punches using a Sunset Laboratory OC-EC analyzer implemented with a thermo-optical analytical method following the EUSAAR-2 protocol (Cavalli et al., 2010). Fourteen (14) crustal and trace elements (Ca, Al, Fe, Ti, V, Ni, Se, As, Sb, Cd, Cr, Mn, Cu, Pb) were analyzed by Inductively Coupled Plasma-Mass Spectrometry (ICP-MS) after acid digestion. For more details on ICP-MS analysis, see Iakovides et al. (2021).

**Quality control:** The quality of ions and carbon measurements was regularly evaluated through inter-laboratory comparison studies coordinated by the Quality Assurance/Science Activity Centre – Americas (QA/SAC-Americas) (http://www.qasac-americas.org) and the European Centre for Aerosol Calibration (ECAC, https://www.actris-ecac.eu/), respectively, while the ICP-MS analytical performance was assessed through inter-laboratory proficiency tests organized by the International Atomic Energy Agency (IAEA).

**Chemical mass closure:** PM mass was reconstructed, as in Bimenyimana et al. (2023), using the chemical species presented above. OM was derived from OC by applying site-specific conversion factors of 1.8 and 2 for the urban site (NICTRA) and the regional background (AMX), respectively (Turpin and Lim, 2001). Mineral dust was determined using the IMPROVE method (Chow et al., 2015; Malm et al., 1994), as shown in Eq. (1), except when trace metal data were not available, in which case it was derived from nss-$Ca^{2+}$ (Sciare et al., 2005).





Mineral dust = 2.2Al + 2.49Si + 1.63Ca + 1.94Ti + 2.42Fe $\qquad$ (1)

The reconstructed PM mass was compared against the gravimetrically determined PM. The two methods exhibited strong agreement, with $r^2$ ranging from 0.84 to 0.99 and slopes > 0.86 (see Fig. S1-A, B and C), further indicating the high quality of our PM chemical composition database.

**Data availability and coverage:** Extensive daily $PM_{10}$ chemical speciation data (3171 and 3356 individual filter samples at NICTRA and AMX, respectively), covering 84-98% of each year, was collected at NICTRA and AMX for the period 2015-2023, and includes all species (ions, carbonaceous species and trace metals). For $PM_{2.5}$, only ions and carbonaceous species were analyzed from 3331 daily filter samples, with a very good annual coverage (82-96%). Daily $PM_{2.5}$ ions data from AMX was also available for the period 2011-2014 and was used only in Sect. 3.6. Further information on the PM chemical speciation data availability at both sites can be found in Table S1.

### 2.4  PM source apportionment

The compiled, extensive PM chemical database (23 individual chemical elements), which comprises a very large number of daily samples accumulated at each site over a decade offers a unique opportunity to perform long-term PM source apportionment and address trends not only for each chemical constituents but also for the various PM sources.

PM source apportionment was carried out using Positive Matrix Factorization (PMF) receptor model (Paatero and Tapper, 1994). The source apportionment analysis was conducted separately for each site using PM chemical composition data covering a wide range of species, including major ions ($Na^+$, $K^+$, $NH_4^+$, $Mg^{2+}$, $Ca^{2+}$, $Cl^-$, $NO_3^-$, and $SO_4^{2-}$), carbonaceous species (OC and EC), several trace metals (Al, Fe, Ti, V, Ni, As, Sb, Cd, Cr, Mn, Cu, and Pb) and PM mass.

Before PMF modelling, missing values (<1% of the total number of samples) were replaced by the geometric mean of measured concentrations for each species, while below detection limit values (generally <10% of the database at both sites, except Cr (44%), Ni (22%) and Cu (21%) at AMX) were substituted by half the method detection limit (MDL) (Polissar et al., 1998). The MDL is defined as the standard deviation of field blank measurements. The uncertainties were estimated as shown in Eq. (2) following Xie & Berkowitz (2006).

$$\sigma_{ij} = \begin{cases} 4*\overline{X_j}, & \text{for missing values,} \\ \dfrac{5}{6}*MDL_{ij}, & \text{for below detection limit values,} \\ \dfrac{MDL_{ij}}{3}+k*X_{ij}, & \text{for measured data,} \end{cases} \qquad (2)$$

where k is a parameter determined by trial and error ranging from 2% to 16%, $X_{ij}$ is the concentration of the $j^{th}$ species in the $i^{th}$ sample, and $MDL_{ij}$ its method detection limit.

### 2.5  Air masses origin and classification

Air masses origin analysis (and their further classification into specific source regions) was performed in order to assign each of the daily PM chemical composition measured at the regional background site (AMX) to a specific source region. This approach allows building a multi-year PM chemical database for each source region; herewith offering the opportunity to assess the influence of these source regions over time.



The Lagrangian particle dispersion model FLEXPART v8.23 (Stohl et al., 2005) was used to trace the origin of air masses
affecting Cyprus at the regional background site (AMX). The model was driven by meteorological input data from National
Center for Atmospheric Research (NCAR) with a spatial resolution of 0.5°x0.5°. For each simulation, forty thousand (40,000)
tracer particles were released every 6 h from the receptor site (AMX) at 350 m AGL and followed 5 days backward in time. A
total of 18,968 retroplumes (an improved substitute for traditional back-trajectories; Stohl et al., 2002) were calculated between
2011 and 2023 and subsequently categorized into seven (7) different source regions (namely North Africa, Middle East,
Europe, West Turkey, Turkey, Marine and Local), based on the Potential Emission Sensitivity (PES), following the same
classification scheme of source regions described by Pikridas et al. (2018).

**2.6 Ancillary observations**

EU regulated ambient trace gases ($NO_x$, $SO_2$ and CO) data was collected alongside PM observations at both NICTRA and
AMX stations throughout the study period. These gases, often co-emitted with primary PM pollutants, can further serve as
specific markers of combustion-related sources such as traffic and industrial emissions. These gaseous pollutant measurements
were conducted using standard techniques namely non-dispersive IR spectroscopy (for CO), chemiluminescence (for $NO_x$),
and UV fluorescence (for $SO_2$) techniques, as described in more details in Vrekoussis et al. (2022).

**3    Results and discussion**

Factors controlling the seasonal variability of $PM_{2.5}$ and $PM_{10}$ at AMX and $PM_{10}$ at NICTRA stations have been extensively
documented by Bimenyimana et al. (2024), along with a comprehensive PM source apportionment and extensive discussion
on the geographic origin of these sources (either local at city scale or regional). Briefly, NICTRA experiences peak PM
concentrations during winter, driven by increased emissions from residential heating and stagnant meteorological conditions
that limit dispersion of atmospheric pollutants. In contrast, higher PM concentrations are recorded during summer at AMX,
due to absence of precipitation (limiting wet scavenging) and enhanced atmospheric (trans)formation processes (Bimenyimana
et al., 2023). These aspects will not be discussed here again. Instead, the following discussion will focus on the inter-annual
variability and long-term trends of the major chemical components and sources.

**3.1  $PM_x$ concentration levels and long-term trends analysis**

Daily values, monthly and annual averages, along with the de-seasonalized monthly means of $PM_x$ mass concentrations for
the urban traffic site (NICTRA) and the regional background (AMX) are shown in Figure 1. Discussions on the trends at both
sites are presented below.

***Regional Background PMx (AMX):*** Annual mean concentrations of 12.4 ± 3.0 µg m⁻³ for $PM_{2.5}$ and 23.9 ± 3.7 µg m⁻³ for
$PM_{10}$ are calculated at AMX from 2005 to 2023. These PM levels are comparable to those observed by Pikridas et al. (2018)
for the period 1998-2015 (14.0 ± 3.6 and 28.7 ± 5.0 µg/m³, for $PM_{2.5}$ and $PM_{10}$, respectively) and Achilleos et al. (2020)
between 2006 and 2017 (25.9 µg/m³ for $PM_{10}$) at the same location.
When considering the entire 19-year (2005-2023) PM database, the Mann-Kendall trend analysis shows significant ($p < 0.01$)
decreasing trends in background $PM_{2.5}$ and $PM_{10}$ levels, with annual rates of -0.31 µg.m⁻³.y⁻¹ (-3.0%.y⁻¹) and -0.49 µg.m⁻³.y⁻¹
(-2.6%.y⁻¹), respectively, corresponding to a total reduction of 31% for $PM_{2.5}$ and 26% for $PM_{10}$, respectively.

These rates are half those previously reported by Pikridas et al. (2018) at the same location (-0.69 µg.m⁻³.y⁻¹ for $PM_{2.5}$ and -
1.1 µg.m⁻³.y⁻¹ for $PM_{10}$) between 2005 and 2015, suggesting a slower decrease in the last decade. This is confirmed when



considering the last 9 years (2015-2023) of our PM dataset which do not show any more a statistically significant decrease in $PM_{2.5}$ and $PM_{10}$ (see Figure 1a and b, and Table 3). These stable $PM_{10}$ levels over the last decade are further discussed in Sect.

3.4 in light of long-term trends in $PM_{10}$ sources.

Interestingly, the observed decreasing trend in $PM_{10}$ concentrations at our regional site (AMX) before 2015 is consistent with the temporal pattern of dust-AOD observed over Cyprus (see Fig. S2) and the broader Mediterranean basin (Logothetis et al., 2021; Marinou et al., 2017), which could suggest that regional dust emissions are an important primary driver of our regional $PM_{10}$ trend. It is also worth noting that the reduction before 2015 in our regional $PM_{2.5}$ (which is largely dominated by

ammonium sulfate, Bimenyimana et al., 2024) is concomitant with the decreasing trend in sulfate levels observed in Europe (Aas et al., 2019, 2024) and the Eastern Mediterranean (Urdiales-Flores et al., 2023) over the last decades which can be attributed to international regulations targeting $SO_2$ emissions (e.g. the Gothenburg Protocol; UNECE, 1999).





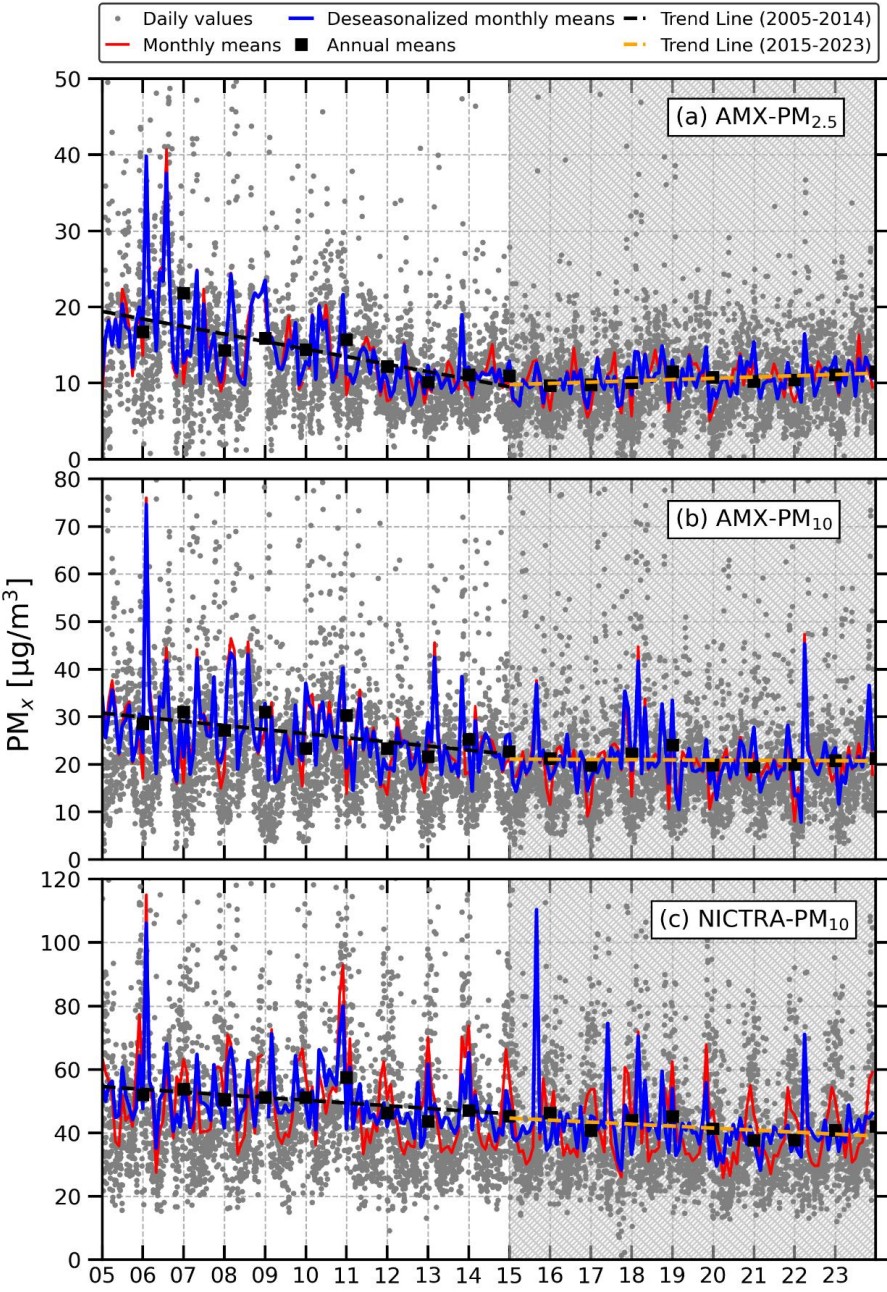

Figure 1. Long-term trends in PM$_{2.5}$ and PM$_{10}$ concentrations from 2005 to 2023. Panels (a) and (b) show the regional background PM$_{2.5}$ and PM$_{10}$ levels (AMX), while panel (c) presents PM$_{10}$ concentrations at the urban traffic site (NICTRA). The grey shaded areas indicate the period (2015-2023) during which complete chemical composition data is available.

**_Urban traffic PM$_{10}$ (NICTRA):_** The urban PM$_{10}$ concentrations (46.0 ± 5.5 µg m$^{-3}$) are twice higher than those observed at

the regional background over the same period (2005-2023), highlighting substantial contribution from local sources from the





urban agglomeration such as traffic emissions, biomass burning, and road dust re-suspension, as extensively discussed by Bimenyimana et al. (2024). These urban PM$_{10}$ concentrations are higher than in most locations of the Central and Western parts of the Mediterranean basin (Conte et al., 2020; Diapouli et al., 2017a; Merico et al., 2019; Pandolfi et al., 2020).

As shown in Figure 1c, urban PM$_{10}$ concentrations show a notable overall reduction of 19% (-0.77 µg.m$^{-3}$.y$^{-1}$; p < 0.001) from 2005 to 2023. This decline is less pronounced than that reported by Pikridas et al. (2018), suggesting, like the regional background site, slower or no improvement in PM$_{10}$ levels in the recent years (from 2015 onward). This is further confirmed by the lack of statistically significant trend for PM$_{10}$ at NICTRA over the period 2015-2023 (-0.24 µg.m$^{-3}$.y$^{-1}$; p >0.1; see Table 3).

The use of the "Lenschow approach" (Lenschow et al., 2001) has been successfully tested and validated in Cyprus for the regional background station (AMX) and the Nicosia urban traffic (NICTRA) (Bimenyimana et al., 2024; Pikridas et al., 2018). This approach assumes that PM$_{10}$ at NICTRA can be decomposed as the sum of a regional PM$_{10}$ component (PM$_{10}$-AMX) and a local PM$_{10}$ fraction (PM$_{10}$-Nicosia) (from emissions within Nicosia). This approach allows to derive a new PM$_{10}$ daily dataset (PM$_{10}$-Nicosia), covering the 19-year period (2005-2023), isolating the contribution from local urban emissions, which

accounts for 50% of total PM$_{10}$ measured at NICTRA.

This results in a calculated decreasing trend of -0.29 µg.m$^{-3}$.y$^{-1}$ (p<0.001) for local PM$_{10}$-Nicosia which can be attributed to a decrease in Nicosia emissions. In other words, local sources are only responsible for 38% of the observed decrease over the period 2005-2023. This suggests that regional emissions have been the major driver of the observed decrease of PM$_{10}$ at our traffic site, which result is quite unexpected given the major contribution of local emissions.

**3.2 Long-term trends in PM chemical composition (2015-2023)**

*Regional background PM$_{2.5}$ (AMX):* The main PM chemical components determined at AMX were averaged over the period 2015-2023. Ammonium sulfate dominates the PM$_{2.5}$ measured at the regional background (AMX) (Figure 2a), making up 43% of total PM$_{2.5}$ mass (up to 50 % during summer), followed by Organic Matter (OM, 29%), and dust (17%). These results are fully consistent with those reported by Bimenyimana et al. (2024) from a one-year study conducted between mid-2016 and

mid-2017 at AMX (48%, 27%, and 13%, respectively, for ammonium sulfate, OM, and dust).

Most of the major PM$_{2.5}$ constituents remained stable (p > 0.1) throughout the study period (2015-2023) (see Table 1 and Fig. S3), a pattern also recorded for PM$_{2.5}$ mass concentrations (see section above). The only exception is K$^+$, a well-known biomass burning tracer, with major influence over the Eastern Mediterranean (e.g. Sciare et al., 2008). This compound shows an increasing trend of +3 ng.m$^{-3}$.y$^{-1}$ (+3.2%.y$^{-1}$; p < 0.001), suggesting a rise in regional biomass burning emissions over the last

decade (2015-2023). However, this increase does not appear to significantly influence the trend in OM which is the main chemical component from biomass burning. Further discussions on this trend are reported in Sect. 3.6 which examines the long-term evolution of this species across different geographic source regions.

*Regional Background PM$_{10}$ (AMX):* As expected, dust constitutes an important fraction of PM$_{10}$ at the regional background (AMX), accounting for 42% of the PM$_{10}$ mass (Figure 2b) and therefore can be considered as a driving force in the inter-

annual variability and trend of PM$_{10}$ in the Eastern Mediterranean (Achilleos et al., 2020). The calculated annual mean PM$_{10}$ dust levels (8.0 µg m$^{-3}$) observed at AMX (2015-2023) further confirm previous findings that Cyprus is the EU country being the most impacted by natural desert dust (Alastuey et al., 2016; Putaud et al., 2010; Querol et al., 2009), which results from substantial influence of dust from the Arabian desert in addition to the Saharan dust storms contribution (Pey et al., 2013; Baars et al., 2016, Achilleos et al., 2020).





As shown in Table 1 and Fig. S4, no statistically significant trends observed for most of the main $PM_{10}$ components for the period 2015-2023, except $K^+$ and OM exhibiting increasing trends of +4 ng.m$^{-3}$.y$^{-1}$ (+3.6%.y$^{-1}$; $p < 0.001$) and +50 ng.m$^{-3}$.y$^{-1}$ (+1.4%.y$^{-1}$; $p < 0.05$), respectively, that could be attributed to regional biomass burning.

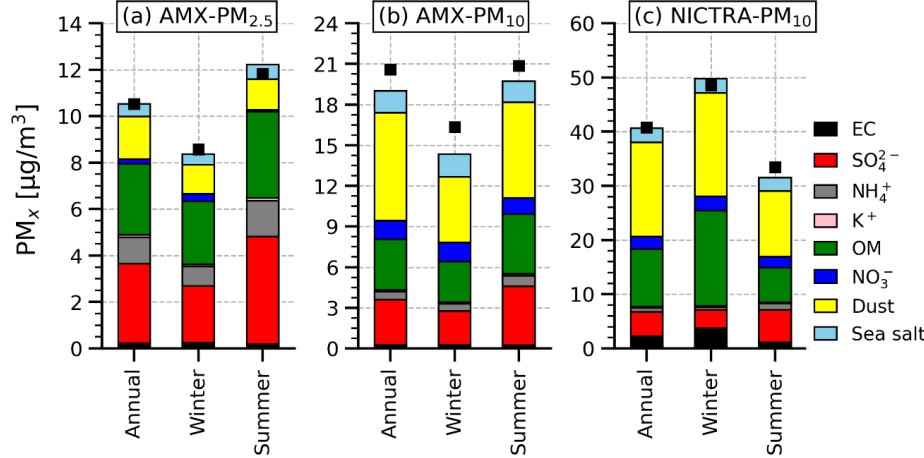

Figure 2. PM chemical composition at AMX and NICTRA for the period 2015-2023. The gravimetric PM masses are shown with black squares.

***Urban traffic $PM_{10}$ (NICTRA):*** As shown in Figure 2c, the urban $PM_{10}$ is primarily composed by dust (43%) and carbonaceous
aerosols (31%, up to 43% during winter). As a matter of fact, their role cannot be overlooked when interpreting the observed multi-annual trends in $PM_{10}$ concentrations.

Long-term trend of PM species at NICTRA is shown in Table 1 and Fig. S5. Similar to the observations made at the regional background site, most key $PM_{10}$ species at NICTRA exhibit stable concentrations for the period 2015-2023, consistently with $PM_{10}$ concentrations reported previously. However, EC shows a significant decline of -0.05 μg.m$^{-3}$.y$^{-1}$ (-2.6%.y$^{-1}$; $p < 0.001$),
most likely due to reductions in local traffic emissions (see Sect. 3.4). Trend analysis was also performed on the local contributions to different PM constituents, derived by applying the "Lenschow" approach. Except OM for which a decreasing trend (-0.11 μg.m$^{-3}$.y$^{-1}$, -2.4%.y$^{-1}$; $p < 0.01$) becomes evident only after excluding the regional fraction, trends for the other species remain almost unchanged.

*Table 1. Mann-Kendall trends analysis (Sen's slope in μg.m$^{-3}$.y$^{-1}$) for PM mass and its major constituents from 2015 to 2023. The stars indicate the level of significance: \*\*\*p < 0.001; \*\*p < 0.01; \*p < 0.05; NS (not significant: p > 0.1).*

|  | Mass | Dust | $SO_4^{2-}$ | OM | EC | $K^+$ | $NO_3^-$ | Sea salt |
|---|---|---|---|---|---|---|---|---|
| AMX-$PM_{2.5}$ | +0.12 (NS) | +0.03 (NS) | -0.02 (NS) | +0.01 (NS) | 0 (NS) | +0.003** | +0.01** | 0 (NS) |
| AMX-$PM_{10}$ | +0.11 (NS) | +0.10 (NS) | -0.01 (NS) | +0.05* | -0.01 (NS) | +0.004*** | -0.01 (NS) | -0.03 (NS) |
| NICTRA-$PM_{10}$ | -0.24 (NS) | +0.13 (NS) | -0.02 (NS) | -0.04 (NS) | -0.05*** | -0.0005 (NS) | +0.03* | -0.01 (NS) |



### 3.3 Overview of the PM source apportionment results over the period 2015-2023

This section presents briefly the main $PM_{10}$ sources at NICTRA and AMX stations, while Sect. 3.4 provides a long-term perspective for each of them.

The PMF optimal solutions for the nine-years period (2015-2023) consist of seven factors for the urban traffic station (NICTRA) and six for the regional background (AMX). Among these sources, five are common to both sites: long range transport (LRT), dust, heavy oil, fresh sea salt and aged sea salt. Biomass burning source and traffic are resolved only at NICTRA, while the regional fossil fuel combustion source is identified only at AMX. The source profiles of the resolved factors are presented in Fig. S6. The PMF solutions obtained here are fully consistent with those reported for both AMX and NICTRA for a 1-year period (Bimenyimana et al., 2024). However, unlike the earlier study, we were able to resolve two distinct sea salt factors (aged and fresh sea salt). The averaged PM source contributions are presented in Table 2, while their multi-annual variability is shown Fig. S7 and Fig. S8.

**Sources identified at both sites (NICTRA and AMX)**

***Dust:*** the mineral dust factor is characterized by high abundance of crustal elements, namely Al (65-82%), Ti (54-79%), Fe (36-78%), $Ca^{2+}$ (40-67%) and Mn (41-73%). It is the most important $PM_{10}$ source (Table 2), with average concentrations of 6.5 and 10.7 µg $m^{-3}$ (33.5 and 27% of $PM_{10}$ mass) at AMX and NICTRA, respectively. There is an additional urban (Nicosia) contribution to this dust factor (dust concentrations in $PM_{10}$ are 65% higher at NICTRA compared to the background levels at AMX). This urban dust is mainly associated with road dust re-suspension as evidenced by the pronounced weekly variability of dust in $PM_{10}$ (NICTRA), with significantly lower ($p < 0.05$) concentrations during weekends (when traffic is expected to be lower) compared to weekdays.

***Long range transport (LRT):*** the LRT factor is found at both stations and exhibits elevated abundance of $NH_4^+$ (83-90%) and $SO_4^{2-}$ (48-49%), species which have a clear regional origin (Bimenyimana et al., 2024). It accounts for 11 and 17% of the $PM_{10}$ mass at NICTRA and AMX, respectively.

***Fresh and aged sea salt:*** $Cl^-$, $Na^+$ and to a lesser extent $Mg^{2+}$ are the major species characterizing the fresh sea salt factor. The $Cl^-/Na^+$ ratios (1.68 and 1.98 for NICTRA and AMX, respectively) are close to the sea water composition (1.8; Seinfeld & Pandis, 2016), suggesting minimally processed (i.e. fresh) sea salt particles. The "fresh sea salt" factor accounts for 5 and 6% of the $PM_{10}$ mass at AMX and NICTRA, respectively.

The "aged sea salt" factor exhibits significant loading of $NO_3^-$ and $SO_4^{2-}$, in addition to $Na^+$. $Cl^-$ is absent in this factor likely due to its depletion as HCl, which is formed through reactions of atmospheric acids ($SO_2$, $HNO_3$) onto sea salt particles. As a factor highly influenced by anthropogenic emissions, aged sea salt concentrations are twice as high as those of fresh sea salt, contributing 11 and 18% to $PM_{10}$ at NICTRA and AMX, respectively (Table 2).

***Heavy oil combustion:*** this factor is characterized by high loading of specific tracers of this source (Celo et al., 2015; Viana et al., 2008), namely V (58-79%) and Ni (61-64%), and is responsible for 5.5 and 7.5% of $PM_{10}$ mass at NICTRA and AMX, respectively.

**Sources resolved only at individual sites (NICTRA or AMX)**

***Traffic*** **(at NICTRA)*:*** the traffic factor is identified by high loading of EC and OC, (ca. 60 and 40% of their respective masses), with an average OC/EC ratio of 1.6 characteristic of tail-pipe vehicular emissions. The high abundance of Cu (79%), Cr (56%), and Fe (46%) suggests emissions from brake and tire abrasion. Traffic is the second most important $PM_{10}$ source at NICTRA, with an overall average contribution of 8.2 µg $m^{-3}$, accounting for 21% of the $PM_{10}$ mass, over the period 2015-2023.



*Biomass burning* **(at NICTRA)***:* the biomass burning factor is found only at NICTRA and consists of a significant abundance
of $K^+$ (40%) and carbonaceous species (40 and 30% of OC and EC total mass, respectively) with a higher OC/EC ratio (3.1)
compared to that of traffic. With an averaged contribution of 19%, biomass burning is almost equivalent to the traffic source
in $PM_{10}$ (at NICTRA) and relates to domestic heating (see Christodoulou et al., 2023; Bimenyimana et al., 2023).

*Regional fossil fuel combustion* **(at AMX)***:* this factor is characterized by carbonaceous species (43 and 58% for OC and EC,
respectively), $NO_3^-$ (32%) and Pb (40%). Interestingly, Pb is not significant in the traffic factor of NICTRA while it is
prominent in this "regional fossil fuel combustion" factor at AMX, likely reflecting the influence of non-EU regulated regional
(Middle East) lead-containing oil. This factor is also associated with elevated OC/EC ratios (average: 5.4), much higher than
the 1.6 found for the freshly emitted traffic factor at NICTRA, supporting a long-range transport origin along with atmospheric
ageing processes. Regional fossil fuel combustion exhibits significant contribution (16%) over the past nine years at AMX
(2015-2023).

Table 2. Average $PM_{10}$ concentrations emitted from different sources ($\mu g\ m^{-3}$) from 2015 to 2023 at NICTRA and AMX, along
with their relative contributions (%) to the PM mass (in parentheses).

|  | Dust | Traffic | Biomass burning | Long range transport | Aged sea salt | Fresh sea salt | Heavy oil | Regional fossil fuel combustion |
|---|---|---|---|---|---|---|---|---|
| NICTRA | 10.7(27) | 8.2(21) | 7.7 (19) | 4.5 (11) | 4.2(11) | 2.2(6) | 2.2(5) | - |
| AMX | 6.9 (35) | - | - | 3.7 (19) | 3.5(18) | 1.3(6) | 1.2(6) | 3.2  (16) |

### 3.4  Trends in PM source contributions

Long-term trends in $PM_{10}$ source concentrations at the urban traffic site (NICTRA) and the regional background (AMX) for
the period 2015-2023 are presented in Table 3, as well as in Fig. S7 and Fig. S8. As mentioned earlier, this period exhibits no
statistically significant trend (increase or decrease) for $PM_{10}$ at either site.

*Traffic emissions (NICTRA):* As shown in Figure 3a and Table 3, a clear downward trend (-0.53 $\mu g.m^{-3}.y^{-1}$ ; $p < 0.001$), is
observed at the urban traffic site (NICTRA) for the road traffic $PM_{10}$ source. Over the past nine years (2015-2023), $PM_{10}$ from
traffic emissions levels have dropped by 35%, from an annual average of 10.1 $\mu g.m^{-3}$ (28% of PM mass) in 2015 to 6.6 $\mu g.m^{-3}$ (16 % of PM mass) in 2023. The major drop in the year 2020 can be attributed to the influence of the lockdown on traffic
emissions in Nicosia as depicted by Putaud et al. (2023).

Similar long-term decreasing trends in traffic emissions have been reported at several locations in Europe (Borlaza et al., 2022;
Diapouli et al., 2017b; In't Veld et al., 2021; Li et al., 2018; Pandolfi et al., 2016). Borlaza et al., (2022) observed a 58%
reduction in $PM_{10}$ from traffic emissions at a remote site in France between 2012 and 2020. Similarly, a 56% decrease for a
mixed industrial/traffic $PM_{2.5}$ factor was reported in Barcelona, Spain by Pandolfi et al. (2016) for the period 2004-2014. For
the Eastern Mediterranean, Diapouli et al. (2017b) observed a drop in $PM_{10}$ traffic exhaust emissions of ca. 45 % for 2011-
2012 compared to 2002 in Athens (Greece). All these results illustrate the efforts to reduce air pollution across Europe through
the implementation of stricter vehicle emission (EURO) standards.

The inter-annual variability of the PM traffic source observed at NICTRA (Figure 3) aligns with the trends in co-located $NO_x$
(exhaust) and Cu (non-exhaust) concentrations, further supporting the decrease of this source at the traffic site. These





observations are also consistent with a 33% reduction in road transport-related BC emissions, as reported in the national

emissions inventory to the European Commission (https://cdr.eionet.europa.eu/cy/un/clrtap/inventories/).

The impact of vehicle fleet modernization on PM pollution has been highlighted by Zhou et al. (2020), who reported that completely phasing out Euro III vehicles and replacing them with cleaner technology like Euro VI could achieve a 99% reduction in PM emissions.

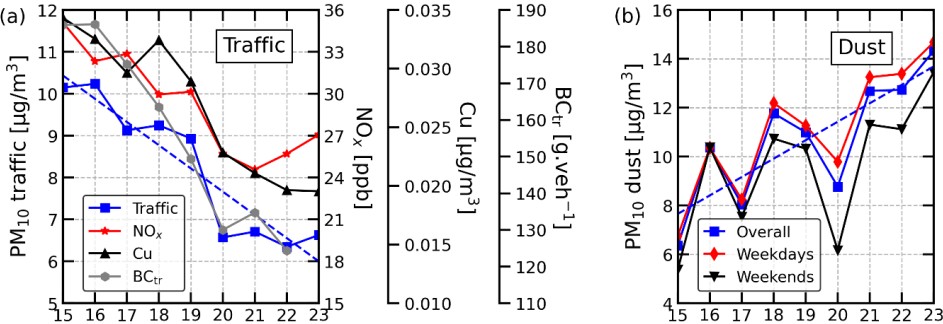

Figure 3. Annual means concentrations of: (a) traffic-related $PM_{10}$ at NICTRA, $NO_x$, Cu along with BC emissions from road transport ($BC_{tr}$; in g.veh$^{-1}$) retrieved from the national emissions inventories, and (b) $PM_{10}$ dust at NICTRA for the period 2015-2023. The blue dashed lines depict the trend lines.

***Dust trends (AMX and NICTRA):*** Mineral dust levels exhibit rapid upward trend at NICTRA (see Figure 3b), with an annual

growth rate of +0.79 µg.m$^{-3}$.y$^{-1}$ (p < 0.001) (see Table 3), offsetting completely the reduction in $PM_{10}$ concentrations resulting from cuts in traffic emissions. This sharp increasing trend is likely to be primarily driven by local road dust emissions. This is supported by the smaller increase in background dust levels at AMX over the same period (+0.19 µg.m$^{-3}$.y$^{-1}$, p<0.1; Table 3) and relatively faster increase during weekdays (+0.80 µg.m$^{-3}$.y$^{-1}$, p<0.01) compared to weekends (+0.63 µg.m$^{-3}$.y$^{-1}$, p<0.01).

***Biomass burning (NICTRA):*** An annual increase in biomass burning emissions of +0.27 µg.m$^{-3}$.y$^{-1}$ (p < 0.001) is observed at

NICTRA (Table 3). This is likely a result of a combination of economic crisis occurred in Cyprus in 2012-2013 and the shift toward cheaper fuels, such as wood, for residential heating in response to rising fuel prices. Such shift was clearly documented in Greece experiencing similar financial crisis during the 2010-2013 period (Fourtziou et al., 2017; Vrekoussis et al., 2013).

***Long range transport (LRT) (AMX and NICTRA):*** As shown in Table 3, the LRT source remains stable over the years for both the urban (NICTRA) and regional background (AMX) stations. However, statistically significant increasing trend is

observed during winter (DJF), with growth rates of +0.24 and +0.17 µg.m$^{-3}$.y$^{-1}$ (p < 0.05), for NICTRA and AMX, respectively. The seasonal switch in air masses origin with a substantial influence of Middle East emissions during winter could explain this pattern and will be further discussed in the next section.







Table 3. Long-term trends in PM$_{10}$ concentrations from different sources at the urban traffic site (NICTRA) and the regional background (AMX) for the period 2015-2023. NS: Not significant.

| Site | Sources | Sen's slope [µg.m$^{-3}$.y$^{-1}$] | Level of significance |
|------|---------|-----------------------------------|----------------------|
| NICTRA | Mineral dust | +0.79 | p < 0.001 |
| | Traffic | -0.53 | p < 0.001 |
| | Biomass burning | +0.27 | p < 0.001 |
| | Long range transport | +0.06 | p > 0.1 (NS) |
| | Aged sea salt | -0.05 | p > 0.1 (NS) |
| | Fresh sea salt | -0.01 | p > 0.1 (NS) |
| | Heavy oil combustion | -0.05 | p < 0.05 |
| AMX | Mineral dust | +0.19 | p < 0.10 |
| | Regional fossil fuel combustion | +0.09 | p < 0.05 |
| | Long range transport | -0.02 | p > 0.10 (NS) |
| | Aged sea salt | +0.05 | p > 0.10 (NS) |
| | Fresh sea salt | -0.04 | p < 0.05 |
| | Heavy oil combustion | -0.09 | p < 0.05 |

***Regional fossil fuel combustion (AMX)***: A small, but statistically significant increase (+0.09 µg.m$^{-3}$.y$^{-1}$; p < 0.05) is also observed for the regional fossil fuel combustion source at the regional background site, in contrast to the major decrease for the traffic source in Nicosia. Previous studies have highlighted a major influence of fossil fuel emissions from Middle East on PM levels over Cyprus (Bimenyimana et al., 2023; Christodoulou et al., 2023).

***Heavy oil combustion (AMX and NICTRA):*** Overall, significant decreasing trends are observed for the heavy oil combustion
source at both NICTRA and AMX sites, with a sharp 34% decline in 2020 compared to 2019, likely attributed to reductions in regional shipping emissions during the COVID pandemic lockdown period and/or improvement in ships fuel oil quality under the 0.5% global sulfur cap which came into force in 2020 (Van Roy et al., 2023; Sofiev et al., 2018).

### 3.5 Influence of regional hotspots on PM source contributions over Cyprus

This section examines the influence of air mass origins on PM source contributions at our regional background receptor site
(AMX) over a nine-year period (2015-2023). A total of 6 clusters (source regions) were identified based on the air masses back trajectory climatology (Pikridas et al., 2018; Bimenyimana et al., 2023); namely Turkey (32%), West Turkey (24%), Middle East (16%), North Africa (13%), Europe (11%) and Marine (3%), (see more information in Sect. 2.5 and Fig. S9). Local cluster was discarded given its lower frequency of occurrence (1%). Such clustering performed over almost a decade allows to characterize the main PM source hotspots (and their location) contributing the most to PM$_{10}$ over Cyprus as measured
at our regional background site (AMX). This approach allows us to better interpret any increasing or decreasing trends observed over Cyprus at our regional background site (AMX).

The resulted PM$_{10}$ source apportionment for our 6 different clusters is reported in Figure 4. Differences between clusters can be interpreted as follows:

***North Africa versus Middle East clusters:*** The North Africa and Middle East regions exhibit the highest PM$_{10}$ levels (32.2
and 29.1 µg m$^{-3}$, respectively; Figure 4). Although these contributions are quite similar, their sources differ considerably. As



expected, desert dust is the most important $PM_{10}$ source for North Africa, with an average contribution of 15.3 µg m$^{-3}$ (57% of $PM_{10}$ mass). The Middle East sector is also characterized by elevated $PM_{10}$ dust levels (10.2 µg m$^{-3}$ or 39% of $PM_{10}$) from the Arabian desert, but also elevated concentrations of anthropogenic pollutants (e.g. of fossil fuel origin), which are nearly double those of North Africa, and are consistent with the well-documented influence of fossil fuel combustion in the Middle East (e.g. Osipov et al., 2022; Paris et al., 2021; Ukhov et al., 2020).

*West Turkey versus other sectors*: As expected, the West Turkey sector is associated with the highest sulfate-rich pollution (found in the Long-range transport source) compared to the other air mass sectors (including the Turkey sector; see Figure 4), primarily due to coal-fired power plants that are mostly concentrated in the western region (Bimenyimana et al., 2024; Pikridas et al., 2010). Interestingly, despite the quasi-absence of coal-fired power plants in the Middle East, the sulfate-rich pollution levels observed for this region are comparable to those of West Turkey (4.3 versus 4.5 µg m$^{-3}$, Figure 4), likely due to emissions from other fossil fuel (oil) combustion sources.

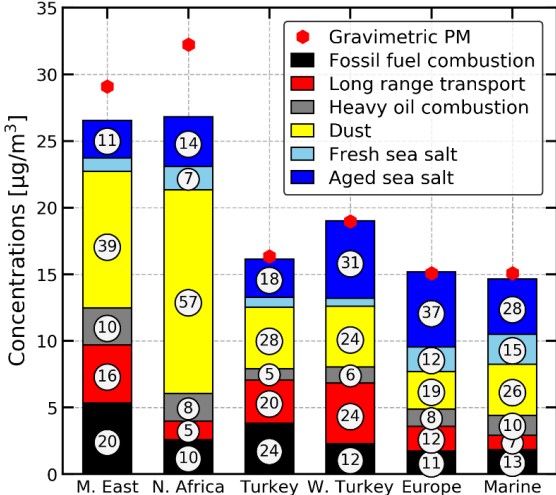

Figure 4. $PM_{10}$ source apportionment for different air masses sectors (Middle East, North Africa, Turkey, West Turkey, Europe, Marine), along with their respective contributions (in percent, to $PM_{10}$ levels (white circles). The red hexagons represent the gravimetric PM mass.

### 3.6 Influence of regional hotspots on long-term PM trends over Cyprus

Within this section we explore the decadal trend of PM chemical composition and sources across various clusters. The daily occurrences for each source area are provided in Table S2 for the period 2011-2023 at the regional background site (AMX). This analysis is made possible thanks to the sufficient number of days for each cluster, except the Marine sector that has limited data and was therefore excluded from further analysis.

**Regional biomass burning emissions**

Because a separate biomass burning factor could not be resolved at the regional background, fine ($PM_{2.5}$) nss-K$^+$ was used as proxy to assess the year-to-year and long-term trends in regional biomass burning emissions. As shown in Figure 5a, the Middle East region is associated not only with the most elevated potassium concentrations but also with a rapid increasing trend (+0.008 µg.m$^{-3}$.y$^{-1}$, p < 0.01), as this species has increased by more than two-fold over the past 13 years (2011-2023)



for this region. Since Middle East air masses are typically observed during winter (see Bimenyimana et al., 2023; Pikridas et al., 2018, and Fig. S9), period during which forest fires are minimal, this rising trend in nss-$K^+$ is most likely due to increasing use of wood for residential heating or increasing agriculture waste burning.

Apart from the Middle East, a rapid upward trend in potassium concentrations is also observed for the West Turkey sector until 2019 (+0.014 $\mu g.m^{-3}.y^{-1}$, $p < 0.05$), followed by a sharp decline afterward (in 2020 and 2021). This increasing is most

probably due to wildfires and/or crop stubble burning, as air masses from this sector are most frequent during summer (see Fig. S9). The drop in 2020-2021 is most likely linked to COVID-19 lockdowns, which have contributed to a reduction in wildfires activity in many regions across the globe (Poulter et al., 2021). The remaining sectors (Turkey, Europe and North Africa) exhibit relatively low and stable potassium levels, with only minimal variations (statistically insignificant) over time.

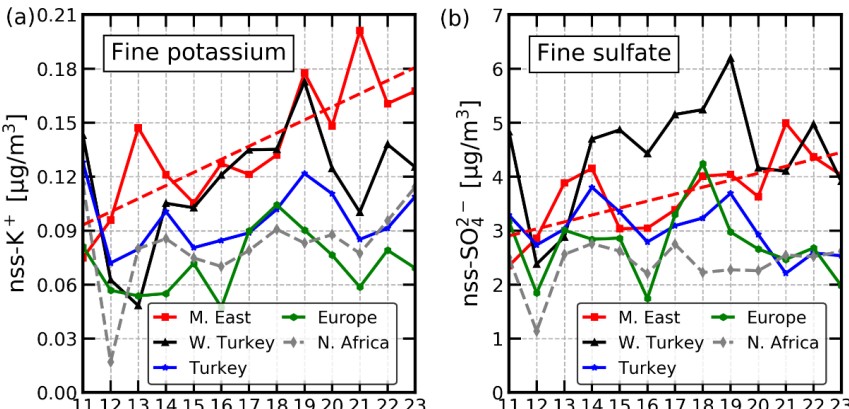

Figure 5. Long-term evolution of: (a) fine nss-$K^+$ and (b) nss-$SO_4^{2-}$ concentrations for different air mass sectors (Middle East, West Turkey, Turkey, Europe, and North Africa) at AMX from 2011 to 2023. The linear regression lines for the Middle East sector are shown with red dashed lines.


**Growing influence of regional sulfur-containing fossil fuel combustion**

Fine ($PM_{2.5}$) sulfate data collected at AMX was used here to describe the influence of the main regional sources of sulfate over time (2011-2023).

As illustrated in Figure 5b, besides having the highest sulfate levels, the West Turkey sector also shows an increasing trend

until 2019 (slope= +0.33 $\mu g.m^{-3}.y^{-1}$, $p < 0.05$), followed by a sharp decrease in 2020 that might be a result of COVID-19 restrictions. This sulfate trend aligns with the growing coal power capacity for Turkey to meet its energy demand (Vardar et al., 2022), despite the efforts to reduce coal use under the Paris agreement. This trend is also consistent with the annual reporting of $SO_2$ emissions from Turkey (EEA, 2023).

Surprising, sulfate concentration levels have been consistently increasing also for the Middle East sector throughout the study

period (2011-2023), with a significant growth rate of +0.14 $\mu g.m^{-3}.y^{-1}$ ($p < 0.01$), rising from 2.2 $\mu g\ m^{-3}$ in 2011 to 4.0 $\mu g\ m^{-3}$ in 2023, an increase of about 82%. This major increase goes along with the upward trend in fossil fuels $CO_2$ emissions observed in the Middle East countries, mainly driven by energy production as well as the oil and gas industry (Friedlingstein et al., 2022; Paris et al., 2021). Similar results are observed for the LRT PMF-resolved factor, which shows a significant increasing trend of +0.54 $\mu g.m^{-3}.y^{-1}$ ($p < 0.05$) for the Middle East sector between 2015 and 2023 (Fig. S10). Interestingly, the Middle East is

gradually surpassing Turkey (the West Turkey sector) which has long been the main origin of sulfate-rich pollution above Cyprus.



## 4 Conclusions

The local (Nicosia) and regional (EMME) long-term trends in PM chemical composition and PM sources were quantified here using a large daily PM chemical composition database collected continuously for a period of nine years (2015–2023) at an urban traffic site of Nicosia (NICTRA) and a regional background site (AMX). Such approach enabled to assess the influence of local and regional emissions on PM long-term variability and to evaluate the potential effectiveness of local abatement measures.

Although $PM_{2.5}$ and $PM_{10}$ concentration levels have decreased at both sites over the last 19 years (2005-2023), concentration levels remain high with no further significant improvements observed over the last 9 years. To further interpret our local and regional PM trends, PMF source apportionment was applied to $PM_{10}$ chemical composition data covering the period 2015-2023. This analysis allowed to apportion seven (7) PM sources at NICTRA and six (6) at AMX, five of which (LRT, dust, heavy oil, fresh sea salt and aged sea salt) being common to both sites, whereas biomass burning and traffic are resolved only at NICTRA and regional fossil fuel combustion at AMX. Dust stands out as the major $PM_{10}$ source at both NICTRA and AMX (27 and 35% of $PM_{10}$ at NICTRA and AMX, respectively), with background $PM_{10}$ levels exceeding those observed at most background/rural locations in Europe due to the proximity of Cyprus to both Saharan and Arabian deserts.

At local scale (Nicosia), a decreasing trend in traffic-related emissions (mostly from exhaust) of 35% (2015-2023) was observed, probably as a result of the gradual switch of the vehicle fleet towards higher EURO standards which are characterized by lower direct (tailpipe) PM emissions. However, these efforts are not reflected in the total urban $PM_{10}$ levels, since uncontrolled emissions from other local sources, such as road dust re-suspension and biomass burning for domestic heating are rising at a rate comparable to that of traffic source, herewith offsetting the benefits induced by the more stringent EURO VI standards.

At regional scale (EMME), the impact of air mass origins on $PM_{10}$ was investigated at the regional background site (AMX) using cluster analysis. While North Africa and Middle East sectors exhibited comparable influence on $PM_{10}$ levels (32.2 versus 29.1 µg m$^{-3}$), the emission sources driving this PM pollution differ significantly between the two sectors. Notably, the Middle East is characterized by not only high dust levels (39%) but also by substantial anthropogenic pollution (nearly double that of North Africa).

This observation is also consistent with the fact that the Middle East sector was found to 1) exhibit an upward trend in sulfate-rich emissions from fossil fuel combustion, gradually exceeding levels observed in Turkey, a country known as one of the largest $SO_2$ emitters worldwide, and 2) increasingly contribute to biomass burning over Cyprus.

Overall, this work suggests that, despite major efforts to reduce PM emissions through adoption of low-emission vehicles, the increase in uncontrolled emissions from local sources (road dust resuspension and biomass burning), along with rising regional fossil fuel emissions mainly from Middle East are likely to undermine efforts to decrease PM concentration levels in Cypriot cities. Consequently it makes it challenging to comply with the stricter $PM_{10}$ limits set by the new EU air quality directives (Directive (EU) 2024/2881; EU, 2024).

**Author contributions**

EB: Acquisition of measurements, data processing and writing-original draft. JS: Funding acquisition, supervision, review, editing and improvement of the manuscript. MP, KO, MI, EV, CS: Acquisition of measurements, data processing, contribution to the original draft. NM: Supervision, review, editing and improvement of the manuscript. All authors have read and agreed to the published version of the manuscript.



**Acknowledgements**

This work has been supported financially by the European Union's Horizon 2020 Research and Innovation Programme and the Cyprus Government under grant agreement no. 856612 (EMME-CARE). The authors sincerely thank the Department of Labor Inspection (DLI) for providing the PM chemical composition data.

**Competing interests**

All authors declare no competing interests.

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
