# Peer review of "Persistent high PM pollution in the Eastern Mediterranean and Middle East: Insights from long-term observations and source apportionment in Cyprus"

_EGUsphere, 2025_

## Referee Comment (RC3)

The authors report a study that performed source apportionment of daily $PM_{10}$ chemical speciation data during 2015–2023 at an urban site (NICTRA) and a rural background site (AMX) in Cyprus in the Eastern Mediterranean and the Middle East region. First, the study found particulate matters levels remained higher than the European Union standard over 2015–2023 based on trend analysis. Second, the study conducted a source apportionment analysis using a positive matrix factorization (PMF) model and identified local and regional for both sites. Among PMF-resolved sources at the urban site, traffic-related emissions decreased, while biomass burning and road dust increased over 2015–2023. Overall, the data are substantial, the method is validated, and the findings are compelling and timely. However, some issues need to be addressed before considering publication in the journal *Atmospheric Chemistry and Physics*. My specific comments are as follows:

Major comments:

1. Insufficient methodological details for PMF. Please provide the rationale for selecting the seven-factor solution for the NICTRA site and six-factor solution for the AMX site as the optimal source factors (Line 262). Specifically, it would be beneficial to include information on the performance of the final PMF solution, including changes in Q value and uncertainty error estimation results such as bootstrapping (BS) and displacement (DISP), if the study used EPA PMF 5.0 software.
2. Missing statistical analysis. Please provide statistical analysis details for the calculation of de-seasonalized monthly average, and for performing non-parametric Mann–kendall test and Sen's slope.
3. Line 152: Please justify the arriving height of 350 m above ground level as the input for running the FLEXPART model at the AMX site, given the AMX is located 532 m above sea level.
4. PMF source assignment issue. For the long-range air mass transport (LRT) source factor, give the high abundance of $NH_4^+$ and $SO_4^{2-}$, this factor should be the secondary sulfate instead.
5. Please consider generating a map showing the air masses back trajectory clusters.
6. Lines 117, 411, and 419: Please clarify the meaning of nss-$Ca^{2+}$, nss-$K^+$, and nss-$SO_4^{2-}$. Please provide details on how these chemical compositions were determined and quantified.
7. The level of significance in non-parametric Mann-Kendall test. What $p$ value does this study consider significant for the Mann-Kendall test? In the Table 1 caption, $p$ value > 0.1 was considered as not significant. and $p$ value < 0.05 was consider as significant, what about the $p$ value range of 0.05–0.1?
8. The study does not provide sufficient discussion on the PMF-resolved PM sources. It would be beneficial to compare the results against existing literature or to evaluate the differences in PM source profiles if the authors consider the manuscript as a research article.

Minor comments:

1. Line 68: Please provide reference(s) for the European Union Air Quality Directive.

2. Line 100: Please spell out the chemical formulas for MSA and KOH upon their first occurrence.
3. Line 114: Please define the abbreviation OM.
4. Line 139: Please clarify the meaning of $\overline{X}_J$. Please double-check the definition of method detection limit, Was the method detection limit determined by three times of the standard deviation above the concentration of blank samples?
5. Line 151: Please specify that NCAR is in the United States.
6. Line 192: Please define the abbreviation dust-AOD.
7. Line 195: The word "concomitant" should be corrected as "consistent".
8. Lines 129–130: Ranges need an en dash and no spaces between start and end (e.g., 2015–2023).
9. Lines 285–288: Please provide reference(s) for $Cl^-$ depletion.
10. Figures S2 and S10. Please clarify the meaning of the dotted lines in the figure captions.

---

## Referee Comment (RC4)

**Peer review report**

**Paper title: Persistent high PM pollution in the Eastern Mediterranean and Middle East: Insights from long-term observations and source apportionment in Cyprus**

**Summary of Strengths, Weaknesses, and Overall Contribution**

This paper presents a valuable long-term dataset on PM concentrations and chemical composition, which enables in-depth analyses of the origins and sources of pollutants. The objectives are clearly defined, and the dataset is of great value to both the local and global scientific community. However, I suggest several improvements to further enhance the quality of the manuscript. Below, I provide general comments followed by more specific ones for consideration.

**Major comments:**

**- Introduction:** While the introduction provides useful background to understand the study's objectives and conclusions, the information is somewhat scattered across paragraphs. I recommend structuring the section so that each paragraph focuses on a specific topic. For example, paragraphs 3 and 4 both address pollutant concentrations in Greece and could be merged and reorganized for clarity.

**- Materials and Methods:** Although the manuscript refers to previous publications for details about the sampling sites, I recommend adding a figure showing land use in the study region together with the locations of the sampling sites. This would help readers better understand the rationale for site selection as well as their proximity to specific areas. Known PM sources or prevailing wind trajectories could also be marked to provide additional context for interpreting the results.

**- Quality control:** Although references are provided regarding quality control procedures, it is important to report specific values of detection limits and quantification limits for the analyzed elements, ions, and other compounds. These data are critical when evaluating the reported concentrations. I recommend including a supplementary table with this information.

**- Results and Discussion:** Given the identified origins of $PM_{10}$, it would strengthen the discussion to include additional references from other authors describing PM sources and variability in the region. This would help contextualize potential sources not only in the study area but also in surrounding regions. Expanding the discussion in this direction would add valuable depth.

**- Line 243:** How do the authors conclude that these elements are related to regional biomass burning? I suggest adding a reference that specifically addresses the origin of these compounds in the region.

**- Urban traffic $PM_{10}$ (NICTRA):** This section would also benefit from a more detailed discussion, ideally comparing the findings with results from other areas in the region or even from other parts of the world. Such comparisons would make the results more meaningful to the broader scientific community.

**Minor comments:**

**- Line 119:** While understandable, the technically correct notation is $R^2$ rather than $r^2$. Please revise accordingly.

**- Line 160:** The sentence *"These gases, often co-emitted with primary PM pollutants, can further serve as specific markers of combustion-related sources such as traffic and industrial emissions"* requires a supporting reference. Please add one.

**- Line 222:** Ammonium sulfate has been previously referred to by its ionic abbreviation. I suggest maintaining this convention throughout the manuscript for all elements and ions. Alternatively, provide both the abbreviation and the full name at first mention and use only abbreviations thereafter for consistency.

**- Line 223:** Acronyms should be defined only upon first use. Please review the manuscript carefully to ensure consistency across all acronyms.

**- Line 429:** Please replace *"surprising"* with *"surprisingly."*

---

## Author Comment (AC1)

**REFEREE #1**

*General Comments*

*This paper focuses on long-term daily measurements of chemical speciation data at 2 coarser size cuts at both an urban site with traffic and a more regional background site in Cyprus in order to distinguish between local and regional emissions and sources. A key finding is a statistically significant drop in particulate matter concentrations which still exceed EU regulations. This paper also focuses on long term trends of individual sources using PMF and found a decrease in traffic-related emissions due to a shift towards EURO-standard vehicles that was largely negated by increases in road dust and biomass burning. They conclude that this region remains a pollutant hotspot due to contributions of desert dust and anthropogenic pollution as well as increases in biomass burning.*

*The introduction focuses on the effects of PM on human health, glances over climate, and establishes the importance of researching it for public policy. They identify gaps in the literature by describing how scarce long-term data is in these regions and how focus has typically instead been on Greece. For example, the authors describe how the most comprehensive long-term PM trend study in Cyprus lacks chemical data to distinguish factors driving downward trends in local urban and regional PM. The annual trends observed are consistent with the literature and the PMF factors resolved at both sites are well supported by tracer analysis and volume of data. The meat of this study is in the trends of identified PMF sources. Overall, I believe that this body of work is good quality, comprehensive, and novel.*

We would like to thank the reviewer for his/her positive comments. Below a point-by-point response to the comments raised. Reviewer's comments are shown in *ITALIC* and our answers are presented in **BLUE.**

*Specific Comments:*

*The agreement in section 2.2 is very high between the gravimetric PM mass determination and TEOM-FDMS and does not raise concern, but can you please be more specific on the number of samples substituted this way?*

Indeed, very strong agreement is observed between TEOM-FDMS and the gravimetric methods ($R^2 = 0.96$; slope = 1.02). Therefore, the gravimetric $PM_{2.5}$ data was substituted with daily averaged online measurements from a co-located TEOM-FDMS. This substitution was applied to $PM_{2.5}$ daily samples collected between 2011 and 2014, corresponding to a total of ca. 1460 samples, or ca. 20% of the dataset.

*In section 2.3, is it known what causes the differences in site-specific conversion factors?*

The selection of different OC-to-OM conversion factors for NICTRA and AMX was motivated by the contrasting nature of these sites. NICTRA being an urban traffic site, is expected to be dominated by primary organic aerosol from direct vehicular emissions, which are typically less oxidized and therefore associated with lower OM/OC. On the other hand, AMX being a regional background site is likely to be significantly influenced by aged secondary organic aerosols during long-range transport, leading to higher OM/OC ratios. Based on the work of Turpin and Lim, (2001) proposing an OC-to-OM conversion factor of $1.6 \pm 0.2$ for urban environments and $2.1 \pm 0.2$ for remote sites, we have applied a value of 1.8 for NICTRA and 2 for AMX to account for their respective characteristics.

*This paper would benefit from a map used to describe the geographic origin of the different source regions as well as a pie chart of their percent frequency.*

As suggested, a map used to describe the geographic location of the different source regions is added in Supplementary material (Fig. S11), along with the frequency (in %) of air masses from the various sectors.

*This paper would also benefit from a short discussion of the differences in de-seasonalized monthly means and monthly means in section 3.1. I understand it is explained in a previous paper, but it seems odd then to present both results in this paper.*

The non-parametric Mann-Kendall trend test is typically applied to de-seasonalized monthly averages rather than monthly means to prevent recurring seasonal fluctuations from masking long-term trends thereby improving the robustness of trend detection. As suggested, a short sentence was added in the revised manuscript.

*This paper's flow would benefit with a short description on what drives the decrease in regional dust emissions at AMX earlier in section 3.1.*

The decrease in regional dust emissions observed before 2015 is likely driven by changes in regional weather patterns that influence dust activity over the region. Shaheen et al. (2023) reported a negative association between dust-AOD and meteorological parameters such as winter sea-level pressure (SLP) and wind regime between 2010 and 2017, reflecting the suppressing effect of these conditions on regional dust. A short sentence was added to the revised manuscript.

*It would be important to describe the methodology for how the optimal number of PMF factors was chosen for both sites. Has separate work been done on temporal correlation of the factors to tracers and of factor concentration to each other? There is often debate about if PMF is distinguishing individual sources or the same source at multiple stages of aging. As an example, an anti-correlation of fresh to aged sea salt could help distinguish these. What method do you use to describe "significant" differences in chemical composition of your factors, specifically in line 285? Given that PMF factor profiles are fixed I wasn't sure how this was determined.*

The methodology for selecting the optimal PMF solution is provided in Section S1 (Supplementary material). This section provides additional information on various quality control checks including correlations of factors with external tracers, Pearson distance (PD = 1-$R^2$), Standardized identity distance (SID), as well as bootstrap results. PD and SID give an indication on the (dis)similarity between PM source profiles. Based on the values of the two metrics, we can conclude that all our PMF factors are significantly different both in terms of temporal correlation (PD ≥ 0.4) and chemical composition (SID ≥ 0.8) (see Tables S3 and S4).

*It confuses me that there is both a decrease in traffic emissions and decrease in heavy oil combustion, but an increase in regional fossil fuel combustion. How well or poorly temporally correlated are the factors to each other? PMF does not provide perfect separation so I worry this conclusion may be driven by regional fossil fuel combustion correlating with dust as it would seem difficult to me to temporally separate local road dust emissions and vehicle emissions. My other concern is that PMF uses a fixed source profile and if there are changes in European car emissions, could other factors be increasing in contribution to compensate? How does the traffic emission factor change in say the first half and second half of the research campaign?*

First of all, we would like to recall that the regional fossil fuel combustion factor was only resolved at AMX. This factor presents no correlation neither with traffic ($R^2$ = 0.06) nor mineral dust ($R^2$ = 0.04) identified at NICTRA. Therefore, its trend is unlikely to be associated with local traffic or road dust emissions from Nicosia city.

PMF source apportionment was also conducted for the beginning (2015) and the end (2023) of the study period to evaluate potential changes in chemical factor profile for the traffic source. No substantial differences are observed between 2015 and 2023, either in terms of concentrations of major species or relative contributions (%) of various tracers. Specifically, the OC/EC ratios are nearly identical for 2015 and 2023 (1.78 and 1.77, respectively). The two profiles were further compared using the standardized

identity distance (SID), one of the similarity metrics proposed by the Joint Research Centre. The obtained SID value (0.7) indicates that the two source profiles are similar.

**Technical Corrections:**

*This paper overall reads smoothly with minimal typos. I've added some comments suggesting ways sentences can be restructured to improve readability and clarity.*

*Line 85*

*"Following rigorously" to "rigorously following"*

Amended, as suggested.

*Line 87*

*"Then, they were" to "They were then"*

This was taken into account in the revised manuscript.

*Line 147*

*"Allows building" to "builds"*

This has been corrected accordingly.

*Line 147*

*Remove "herewith"*

As suggested, we have removed "herewith".

*Line 162*

*"details" to "detail"*

"details" was replaced by "detail" as suggested by the reviewer.

*Line 340*

*"offsetting completely" to "completely offsetting".*

Changes were made in the revised manuscript, accordingly.

*Line 417*

*"minimal variations (statistically insignificant)" to "minimal and statistically significant variations"*

Here we meant no statistically significant variations. Changes have been made in the revised manuscript.

*Line 424*

*"As illustrated in Figure 5b, besides having the highest sulfate levels, the West Turkey sector also shows an increasing trend" to "As illustrated in Figure 5b, the West Turkey sector has the highest sulfate levels and also shows an increasing trend".*

Given that this paragraph focuses on sulfate trends rather than concentration levels, the sentence was revised as follows: "As illustrated in Figure 5b, the West Turkey sector not only has the highest sulfate levels but also exhibits an increasing trend…."

*Line 429*

*"Surprising" to "Surprisingly"*

We thank the reviewer for pointing out this typo, which has been corrected in the revised manuscript.

*Line 440*

*"Such approach" to "This approach"*

As suggested by the reviewer, "Such approach" was replaced by "This approach".

*Line 447*

*"being common" to "were common"*

We have kept "being common" instead of "were common" in this context because it better fits the flow of the manuscript.

*Line 448*

*"at AMX" to "only at AMX"*

The sentence was revised accordingly.

*Line 461*

*"(nearly double that of North Africa) to "that is nearly double that of North Africa)"*

The sentence was revised following the reviewer's suggestion.

Line 466

"uncontrolled emissions from local sources (road dust resuspension and biomass burning)," to "uncontrolled emissions, road dust resuspension and biomass burning, from local sources,"

We thank the reviewer for this suggestion. However, we have decided to keep this sentence in its initial form because it better fits the flow of the manuscript.

**REFERENCES**

Shaheen, A., Wu, R., Yousefi, R., Wang, F., Ge, Q., Kaskaoutis, D. G., Wang, J., Alpert, P., and Munawar, I.: Spatio-temporal changes of spring-summer dust AOD over the Eastern Mediterranean and the Middle East : Reversal of dust trends and associated meteorological effects, Atmos. Res., 281, 106509, https://doi.org/10.1016/j.atmosres.2022.106509, 2023.

Turpin, B. J. and Lim, H. J.: Species contributions to pm2.5 mass concentrations: Revisiting common assumptions for estimating organic mass, Aerosol Sci. Technol., 35, 602–610, https://doi.org/10.1080/02786820119445, 2001.

---

## Author Comment (AC2)

**REFEREE #2**

*General comment:*

*The paper reports a long-term study of PM concentration and composition at two sites of eastern Mediterranean area. The trends of the contributions of the different sources are investigated by means of source apportionment. The topic is interesting and suitable for the Journal. The paper is clear and generally well written. I just have some minor suggestions detailed in my specific comments.*

We would like to thank the reviewer for his/her positive comments. Below a point-by-point response to the comments raised. Reviewer's comments are shown in *ITALIC* and our answers are presented in **BLUE.**

*Specific comments:*

*Lines 30-35. I would suggest to mention the recent works on trends of composition and sources in a southern Italy background site that could be representative of a central or central-east Mediterranean area such as Merico et al (Atmospheric Pollution Research, 102668, 2025) and Giannossa et al. (Journal of Environmental Management 319, 115752, 2022).*

We thank the reviewer for this suggestion. The proposed references have been added to the revised manuscript.

*Section 2.4. All years have been used as a single input dataset?*

Indeed, the 9-year chemical composition data was used as a single PMF input dataset. This approach improves the stability of PMF factors and enhances the robustness of the results. Moreover, it produces consistent source profiles across the years, allowing us to track the evolution of each source contribution over time, which is the focus of this study. Such an explanation was now added in the manuscript.

*Lines 191-194. Here it seems that there is a decreasing trend on dust coming out from chemical mass closure; however, source apportionment results indicate an increasing trend of this source. Could this aspect be discussed in more detail?*

The decreasing trend of dust (based on dust-AOD) discussed in Lines 191-194 is observed until 2015 but does not persist thereafter. For the period covered by PMF analysis (2015–2023), dust estimated by both methods (chemical mass closure and PMF) exhibits an increasing trend, although the magnitude of this trend is different (see Tables 1 and 3). These differences arise from the fact that only PMF apportions different crustal elements among multiple dust sources, while chemical mass closure groups them all under mineral dust component. For instance, a substantial fraction of Fe was resolved by PMF in traffic factor, reflecting contributions from brake and tire wear.

*Lines 303-308. Regional fossil fuel combustion. I would suggest naming this component just as combustion sources because it is likely a mixed source including both traffic and biomass burning. This will justify the large OC/EC ratio and the aspect that biomass burning is surprisingly identified at traffic site but not a background. This kind of mixing could be due to the lack of a specific tracer for biomass burning such as levoglucosan and has been observed also in other sites.*

While the influence of biomass burning to this factor cannot be ruled out, it is expected to be minimal given that most $K^+$ (a tracer of biomass burning; Fourtziou et al., 2017; Puxbaum et al., 2007) is found in the LRT factor. Therefore, the name "regional fossil fuel combustion" is kept to encompass contributions from fossil fuel sources. The fact that biomass burning is resolved at the urban site

(NICTRA) but not at the regional background is very reasonable given the large emissions from domestic heating during winter period (Bimenyimana et al., 2024; Christodoulou et al., 2023). In contrast, the regional background (AMX) is only affected by episodic forest fire emissions during summer (Bimenyimana et al., 2024). Although levoglucosan has been used to resolve biomass burning source at various sites, it may be inappropriate for our regional background site given its fast atmospheric oxidation, leading to a significant degradation of this species during long range transport (Theodosi et al., 2018).

*The profile named LRT is usually called sulphate or secondary sulphate in PMF-based source apportionment. I would suggest to consider this flag.*

Although our LRT is usually labelled as "sulphate" or "secondary sulphate" in PMF-based source apportionment studies, we propose naming this factor "Regional secondary aerosol" to account for the other secondary species, such as secondary organic aerosols, that it contains and to better reflects its regional origin rather than limiting its interpretation to a single chemical constituent.

**REFERENCES**

Bimenyimana, E., Sciare, J., Oikonomou, K., Iakovides, M., Pikridas, M., Vasiliadou, E., Savvides, C., and Mihalopoulos, N.: Cross-validation of methods for quantifying the contribution of local (urban) and regional sources to PM2. 5 pollution: application in the Eastern Mediterranean (Cyprus)., Atmos. Environ., 120975, 2024.

Christodoulou, A., Stavroulas, I., Vrekoussis, M., Desservettaz, M., Pikridas, M., Bimenyimana, E., Kushta, J., Ivančič, M., Rigler, M., Goloub, P., Oikonomou, K., Sarda-Estève, R., Savvides, C., Afif, C., Mihalopoulos, N., Sauvage, S., and Sciare, J.: Ambient carbonaceous aerosol levels in Cyprus and the role of pollution transport from the Middle East, Atmos. Chem. Phys., 23, 6431–6456, https://doi.org/10.5194/acp-23-6431-2023, 2023.

Fourtziou, L., Liakakou, E., Stavroulas, I., Theodosi, C., Zarmpas, P., Psiloglou, B., Sciare, J., Maggos, T., Bairachtari, K., Bougiatioti, A., and others: Multi-tracer approach to characterize domestic wood burning in Athens (Greece) during wintertime, Atmos. Environ., 148, 89–101, 2017.

Puxbaum, H., Caseiro, A., Sánchez-Ochoa, A., Kasper-Giebl, A., Claeys, M., Gelencsér, A., Legrand, M., Preunkert, S., and Pio, C.: Levoglucosan levels at background sites in Europe for assessing the impact of biomass combustion on the European aerosol background, J. Geophys. Res. Atmos., 112, 2007.

Theodosi, C., Panagiotopoulos, C., Nouara, A., Zarmpas, P., Nicolaou, P., Violaki, K., Kanakidou, M., Sempéré, R., and Mihalopoulos, N.: Sugars in atmospheric aerosols over the Eastern Mediterranean, Prog. Oceanogr., 163, 70–81, https://doi.org/10.1016/J.POCEAN.2017.09.001, 2018.

---

## Author Comment (AC3)

**REFEREE #3**

*The authors report a study that performed source apportionment of daily PM10 chemical speciation data during 2015–2023 at an urban site (NICTRA) and a rural background site (AMX) in Cyprus in the Eastern Mediterranean and the Middle East region. First, the study found particulate matters levels remained higher than the European Union standard over 2015 2023 based on trend analysis. Second, the study conducted a source apportionment analysis using a positive matrix factorization (PMF) model and identified local and regional for both sites. Among PMF-resolved sources at the urban site, traffic-related emissions decreased, while biomass burning and road dust increased over 2015–2023. Overall, the data are substantial, the method is validated, and the findings are compelling and timely. However, some issues need to be addressed before considering publication in the journal Atmospheric Chemistry and Physics.*

We would like to thank the reviewer for his/her positive comments. Below a point-by-point response to the comments raised. Reviewer's comments are shown in *ITALIC* and our answers are presented in **BLUE.**

*My specific comments are as follows:*

*Major comments:*

*1. Insufficient methodological details for PMF. Please provide the rationale for selecting the seven-factor solution for the NICTRA site and six-factor solution for the AMX site as the optimal source factors (Line 262). Specifically, it would be beneficial to include information on the performance of the final PMF solution, including changes in Q value and uncertainty error estimation results such as bootstrapping (BS) and displacement (DISP), if the study used EPA PMF 5.0 software.*

The methodology used to determine the optimal number of factors and several quality control tests are provided in Section S1 (Supplementary material). This section also presents the error estimation results obtained through bootstrapping (Table S5).

*2. Missing statistical analysis. Please provide statistical analysis details for the calculation of de-seasonalized monthly average, and for performing non-parametric Mann–Kendall test and Sen's slope.*

The methodology for the Mann-Kendall test has been added to the revised manuscript.

*3. Line 152: Please justify the arriving height of 350 m above ground level as the input for running the FLEXPART model at the AMX site, given the AMX is located 532 m above sea level.*

The arriving height of 350 m AGL was chosen for FLEXPART simulations to ensure that the retroplumes remain within the typical mixing layer at AMX (PBLH profiles from the ceilometer are available at https://e-profile.eu/), while minimizing potential effects of near-surface processes.

*4. PMF source assignment issue. For the long-range air mass transport (LRT) source factor, give the high abundance of $NH_4^+$ and $SO_4^{2-}$, this factor should be the secondary sulfate instead.*

This has been addressed before: Although our LRT is usually labelled as "sulphate" or "secondary sulphate" in PMF-based source apportionment studies, we propose naming this factor "Regional secondary aerosol" to account for the other secondary species, such as secondary organic aerosols, that it contains and to better reflect its regional origin rather than limiting its interpretation to a single chemical constituent.

*5. Please consider generating a map showing the air masses back trajectory clusters.*

This has been addressed before (see Response to Referee #1): A map used to describe the geographic location of the different source regions is available in Supplementary material (Fig. S11).

*6. Lines 117, 411, and 419: Please clarify the meaning of nss-Ca$^{2+}$, nss-K$^+$, and nss-SO$_4^{2-}$. Please provide details on how these chemical compositions were determined and quantified.*

As suggested by the reviewer, definitions of nss-Ca$^{2+}$, nss-K$^+$, and nss-SO$_4^{2-}$ have been provided in the revised manuscript.

*7. The level of significance in non-parametric Mann-Kendall test. What p value does this study consider significant for the Mann-Kendall test? In the Table 1 caption, p value > 0.1 was considered as not significant. and p value < 0.05 was consider as significant, what about the p value range of 0.05–0.1?*

Statistical tests are generally considered significant for p-value < 0.05 (at 95 % confidence level). Sometimes, p < 0.1 is also considered significant. In the current study, p values were either < 0.05 or > 0.1 (not significant).

*8. The study does not provide sufficient discussion on the PMF-resolved PM sources. It would be beneficial to compare the results against existing literature or to evaluate the differences in PM source profiles if the authors consider the manuscript as a research article.*

A brief discussion comparing the results with existing literature has been added, as suggested by the reviewer.

*__Minor comments:__*

*1. Line 68: Please provide reference(s) for the European Union Air Quality Directive.*

A reference for the EU Air Quality Directive (Directive 2008/50/EC) has been provided.

*2. Line 100: Please spell out the chemical formulas for MSA and KOH upon their first occurrence.*

The full spelling of MSA and KOH has been included in the revised manuscript.

*3. Line 114: Please define the abbreviation OM.*

The definition of OM has been included in the revised manuscript, as suggested by the reviewer

*4. Line 139: Please clarify the meaning of $\overline{X_j}$. Please double-check the definition of method detection limit, Was the method detection limit determined by three times of the standard deviation above the concentration of blank samples?*

The definition of $\overline{X_j}$ was provided in the revised manuscript.

Indeed, the method detection limit was calculated as three times the standard deviation of field blank concentrations. This has been corrected.

*5. Line 151: Please specify that NCAR is in the United States.*

This has been taken into account in the revised manuscript.

*6. Line 192: Please define the abbreviation dust-AOD.*

The definition of dust-AOD has been included.

*7. Line 195: The word "concomitant" should be corrected as "consistent".*

This has been corrected, as suggested by the reviewer.

*8. Lines 129–130: Ranges need an en dash and no spaces between start and end (e.g., 2015–2023).*

This has been considered throughout the revised manuscript.

*9. Lines 285–288: Please provide reference(s) for Cl– depletion.*

A reference for $Cl^-$ has been added, as suggested by the reviewer.

*10. Figures S2 and S10. Please clarify the meaning of the dotted lines in the figure captions.*

This has been taken into consideration.

---

## Author Comment (AC4)

**REFEREE #4**

*Summary of Strengths, Weaknesses, and Overall Contribution*

*This paper presents a valuable long-term dataset on PM concentrations and chemical composition, which enables in-depth analyses of the origins and sources of pollutants. The objectives are clearly defined, and the dataset is of great value to both the local and global scientific community. However, I suggest several improvements to further enhance the quality of the manuscript. Below, I provide general comments followed by more specific ones for consideration.*

We would like to thank the reviewer for his/her positive comments. Below a point-by-point response to the comments raised. Reviewer's comments are shown in *ITALIC* and our answers are presented in **BLUE.**

*Major comments:*

*-Introduction: While the introduction provides useful background to understand the study's objectives and conclusions, the information is somewhat scattered across paragraphs. I recommend structuring the section so that each paragraph focuses on a specific topic. For example, paragraphs 3 and 4 both address pollutant concentrations in Greece and could be merged and reorganized for clarity.*

We thank the reviewer for the comment. The introduction section has been revised accordingly.

*-Materials and Methods: Although the manuscript refers to previous publications for details about the sampling sites, I recommend adding a figure showing land use in the study region together with the locations of the sampling sites. This would help readers better understand the rationale for site selection as well as their proximity to specific areas. Known PM sources or prevailing wind trajectories could also be marked to provide additional context for interpreting the results.*

A figure describing the study area (for both NICTRA and AMX) have been added to the manuscript.

*- Quality control: Although references are provided regarding quality control procedures, it is important to report specific values of detection limits and quantification limits for the analyzed elements, ions, and other compounds. These data are critical when evaluating the reported concentrations. I recommend including a supplementary table with this information.*

As suggested by the reviewer, the method detection limits and quantification limits have been reported in Supplementary material (Table S2).

*- Results and Discussion: Given the identified origins of $PM_{10}$, it would strengthen the discussion to include additional references from other authors describing PM sources and variability in the region. This would help contextualize potential sources not only in the study area but also in surrounding regions. Expanding the discussion in this direction would add valuable depth.*

Relevant references to other studies in the region have been considered, as suggested by the reviewer

*-Line 243: How do the authors conclude that these elements are related to regional biomass burning? I suggest adding a reference that specifically addresses the origin of these compounds in the region.*

$K^+$ (and OM to a lower extent) is usually used as tracer for biomass burning (Fourtziou et al., 2017; Puxbaum et al., 2007; Reche et al., 2012). Given that the influence of local emissions at our regional background site is minimal, this species is likely to originate from regional biomass burning. This is further supported by a peak of $K^+$ concentrations observed during summer most probably due to agricultural waste and/or forest fires (see Bimenyimana et al., 2025). References were now added, as suggested by the reviewer.

*-Urban traffic PM10 (NICTRA):* *This section would also benefit from a more detailed discussion, ideally comparing the findings with results from other areas in the region or even from other parts of the world. Such comparisons would make the results more meaningful to the broader scientific community.*

As suggested by the reviewer, discussion on PM trends at other locations of the Mediterranean (Greece and Spain) has been added.

*Minor comments:*

*-Line 119:* *While understandable, the technically correct notation is $R^2$ rather than $r^2$. Please revise accordingly.*

This has been corrected throughout the manuscript.

*-Line 160:* *The sentence "These gases, often co-emitted with primary PM pollutants, can further serve as specific markers of combustion-related sources such as traffic and industrial emissions" requires a supporting reference. Please add one.*

A reference has been added, as suggested.

*-Line 222:* *Ammonium sulfate has been previously referred to by its ionic abbreviation. I suggest maintaining this convention throughout the manuscript for all elements and ions. Alternatively, provide both the abbreviation and the full name at first mention and use only abbreviations thereafter for consistency.*

This suggestion has been considered in the revised manuscript.

*-Line 223:* *Acronyms should be defined only upon first use. Please review the manuscript carefully to ensure consistency across all acronyms.*

All acronyms are now defined at first use and used consistently, as suggested by the reviewer.

*-Line 429:* *Please replace "surprising" with "surprisingly".*

This has been corrected in the revised manuscript.

**REFERENCES**

Bimenyimana, E., Sciare, J., Oikonomou, K., Iakovides, M., Pikridas, M., Vasiliadou, E., Savvides, C., and Mihalopoulos, N.: Cross-validation of methods for quantifying the contribution of local (urban) and regional sources to PM2.5 pollution: application in the Eastern Mediterranean (Cyprus)., Atmos. Environ., 120975, 2025.

Fourtziou, L., Liakakou, E., Stavroulas, I., Theodosi, C., Zarmpas, P., Psiloglou, B., Sciare, J., Maggos, T., Bairachtari, K., Bougiatioti, A., and others: Multi-tracer approach to characterize domestic wood burning in Athens (Greece) during wintertime, Atmos. Environ., 148, 89–101, 2017.

Puxbaum, H., Caseiro, A., Sánchez-Ochoa, A., Kasper-Giebl, A., Claeys, M., Gelencsér, A., Legrand, M., Preunkert, S., and Pio, C.: Levoglucosan levels at background sites in Europe for assessing the impact of biomass combustion on the European aerosol background, J. Geophys. Res. Atmos., 112, 2007.

Reche, C., Viana, M., Amato, F., Alastuey, A., Moreno, T., Hillamo, R., Teinilä, K., Saarnio, K., Seco, R., Peñuelas, J., Mohr, C., Prévôt, A. S. H., and Querol, X.: Biomass burning contributions to urban aerosols in a coastal Mediterranean City, Sci. Total Environ., 427–428, 175–190,

https://doi.org/10.1016/j.scitotenv.2012.04.012, 2012.